# Early Warning Score Trend Analysis: A Data-Driven Approach for Emergency Medical Services

1st Tamara Krafft
*Faculty of Applied Computer Science*
*University of Augsburg*
Augsburg, Germany
tamara.krafft@uni-a.de

2nd Fabian Stieler
*Faculty of Applied Computer Science*
*University of Augsburg*
Augsburg, Germany
fabian.stieler@uni-a.de

3rd Bernhard Bauer
*Faculty of Applied Computer Science*
*University of Augsburg*
Augsburg, Germany
bernhard.bauer@uni-a.de

*Abstract*—Early recognition of clinical deterioration is crucial for timely intervention, especially during Emergency Medical Services (EMS) encounters. Early Warning Scores (EWS) translate raw vital signs into a clinically transparent risk scale. However, research on prehospital EWS applications is limited and often focuses on in-hospital outcomes and single snapshots, neglecting short-term risk trajectories. This paper explores whether EWS trends, captured just before initial EMS intervention, convey additional information and can predict the return of spontaneous circulation (ROSC) during out-of-hospital cardiac arrest encounters. In a retrospective study of 4,394 cardiac arrest encounters from the 2021–2023 National EMS Information System (NEMSIS), we applied eight different EWS models at every documented vital sign measurement and derived time-normalized preintervention features, including slope, mean, area under the EWS curve (AUC), and exponentially weighted average (EWA). Informational value was quantified with nonparametric tests and L1-regularized logistic regression models targeting prehospital ROSC. Our findings demonstrate that short-term EWS dynamics encode measurable patterns of clinical deterioration, achieving moderate predictive performance (AUROC: 0.665) and advancing the current understanding of prehospital risk assessment. These results highlight the potential of incorporating vital sign trajectories into real-time, data-driven decision-support tools for EMS and motivate further exploration with more flexible, AI-based modeling approaches.

*Index Terms*—Emergency Medical Services (EMS), Prehospital Care, Risk Stratification, Early Warning Scores (EWS), Clinical Decision Support, Machine Learning

## I. Introduction

Acute health conditions are frequently preceded by measurable changes in vital signs [1]. Early recognition of progressive physiological decline in a patient, known as clinical deterioration [2], is critical for the initiation of timely and effective interventions and has been associated with improved patient outcomes and reduced mortality rates [3], [4]. For instance, timely interventions on patients experiencing ventricular fibrillation prior to out-of-hospital cardiac arrest have been shown to enhance survival rates significantly [5].

Hospitals widely employ Early Warning Score (EWS) models, such as the National Early Warning Score (NEWS) [6], as a standard procedure to support early detection. These models function as aggregated track-and-trigger systems, assigning scores to routinely measured vital signs, including heart rate, respiratory rate, and systolic blood pressure, based on clinically validated thresholds [7]. Ultimately, this results in a clinically transparent indicator of a patient's risk of deterioration, facilitating risk stratification and the initiation of appropriate care [7].

Although EWS models have demonstrated success in hospital environments, their integration into prehospital care remains limited. Previous studies have demonstrated the potential utility of EWS in Emergency Medical Services (EMS) contexts [8]–[11]. However, most studies focus on longer-term outcomes, such as in-hospital mortality or ICU admission [8]–[11], offering limited insight into on-scene patient trajectories. Moreover, EWS models are typically applied as static snapshots in time, although continuously monitored vital signs may be available. Veldhuis et al. [12] found that upward trends in emergency department (ED) MEWS correlate with the onset of critical illness, suggesting that temporal data holds additional informational value. In prehospital environments, relying on established in-hospital risk thresholds may further limit EWS applicability. Guan et al. [11] note that thresholds may not generalize well due to differing patient characteristics in EMS.

Thus, this paper investigates the temporal patterns of vital signs in the moments leading up to initial EMS intervention. Specifically, we evaluate whether EWS models, used as clinically transparent risk proxies for vital sign data, can capture clinical deterioration before the initiation of intervention. Furthermore, this paper explores the statistical separability and predictive performance of various EWS-derived temporal features concerning documented return of spontaneous circulation (ROSC) prior to hospital arrival.

The objective of this study is to validate the informational value of preintervention vital sign monitoring and temporal modeling to establish a foundation for future artificial intelligence (AI)-driven decision support tools in EMS, which leverage short-term deterioration patterns. Based on our findings, we outline and discuss opportunities and remaining challenges for integrating AI methods into EMS.

The rest of the paper is structured as follows. First, Section II introduces data and methodology. Next, Section III reports our findings on preintervention trends, feature separability, and predictive performance. Section IV interprets these results in a broader context and acknowledges limitations. Finally, Section V concludes the paper and sketches out possible future directions.

## II. METHODS

We conducted a retrospective analysis of three consecutive years (2021-2023) of data from the National EMS Information System (NEMSIS) [13]. The methodology consists of four parts: vital sign extraction, EWS scoring, feature extraction, and predictive analysis. We extracted the vital sign set for a fixed number of data points preceding specific interventions from multiple EMS encounters. At each data point, we computed the scores of eight different EWS models and derived a range of features for each encounter. These features were then analyzed to assess their correlation with clinical deterioration and their predictive value for determining the occurrence of ROSC prior to arrival at the ED.

### A. Data

The NEMSIS database is a national EMS registry that captures de-identified, standardized electronic care reports submitted by U.S. EMS agencies, offering granular vital sign data of more than 49 million EMS activations per year.

Our analysis focused on encounters involving cardiac arrest, specifically those in which external ventricular defibrillation or cardiopulmonary resuscitation (CPR) was performed. These procedures were selected to focus on clearly defined interventions, as NEMSIS categorizes a broad range of procedures, including routine activities, such as patient repositioning. We extracted four key vital signs: heart rate, respiratory rate, systolic blood pressure, and oxygen saturation. We applied forward filling to address missing vital sign data. This approach avoids assuming future values, which is an essential consideration for real-time applicability. Subsequently, encounters with fewer than five complete vital sign sets preceding the first qualifying intervention were excluded from further analysis. Furthermore, encounters in which the five-point window exceeded 20 minutes were excluded to keep the observation horizon clinically realistic. Finally, we utilized the documented ROSC outcome as the outcome variable in our analysis of the association between vital sign trends and patient trajectory.

Following all filtering steps, the final dataset contains 4,394 encounters: 1,279 from 2021, 1,471 from 2022, and 1,644 from 2023. Of these encounters, 1,519 had a documented ROSC prior to ED arrival, and 2,875 had no ROSC. The regarded five-point window had a median length of 840 seconds (IQR: 608-1001). The Supplementary Material[1] details cohort characteristics and code snippets that demonstrate the data extraction directly from the public-release NEMSIS data.

[1] https://osf.io/q7pfw/?view_only=bd9e534918ed4eb59309fe9718e7ea58

### B. Early Warning Score models

We employed the EWS concept as a clinically transparent proxy for overall patient health status. EWS models convert vital sign measurements into aggregated scores using vital-specific thresholds, enabling standardized assessment of deterioration risk. In our work, we applied multiple EWS models to reduce potential bias, as different models may demonstrate variable performance [9]. We determined the final set of EWS models used in this study based on Gerry et al. [7], who identified 84 EWS validation studies applied to adult hospital patients, across which 22 distinct EWS models were evaluated. For our study, a subset of these models was selected by excluding models that required laboratory values or more than three parameters beyond the four previously selected vital signs. Disease-specific EWS models, such as qSOFA, and models with proven poor discriminatory power [14], such as the EWS proposed by Groarke et al., were omitted.

Ultimately, eight EWS models were selected: NEWS, MEWS, ViEWS, CEWS, SEWS, Worthing PSS, RAPS, and GMEWS. Parameters for which data were unavailable (body temperature, mean arterial pressure and level of consciousness) were omitted. All remaining thresholds were applied unchanged at each timestamp to compute each model's aggregate total. Crucially, we do not map these totals to in-hospital risk cut-offs, as omitting parameters lowers the maximum attainable score and invalidates cut-offs. Supplementary Material[1] lists all eight models, their parameter thresholds, and a computation example.

### C. Derived Features

To assess whether preintervention trajectories encode clinically relevant information, we defined a fixed analysis window consisting of the five most recent measurements preceding the selected intervention event. The EWS trajectory of a selected NEMSIS encounter is illustrated in Figure 1 along with a subset of the derived features. We ran a sensitivity analysis[1] varying the window from three to ten measurements to verify this choice, which confirmed that a five-point window offers the best trade-off between model performance and cohort size. Because prehospital measurements are prone to occur at irregular intervals, all derived features were subsequently time-normalized to ensure comparability across differing temporal spans.

We estimated the slope of the EWS trend per window using a simple linear function, $\text{EWS}(t) = mt + b$, where $t$ denotes the actual timestamp of each measurement and the slope $m$ captures the linear trend of EWS values over time. A positive $m$ indicates a consistent increase in EWS scores, potentially signaling clinical deterioration. Additionally, we fit a quadratic function to the EWS values over the same window to capture non-linear trends, $\text{EWS}(t) = at^2 + bt + c$. As a measure of curvature, we extracted the second derivative of this function, given by $2a$, which reflects the rate of change in the slope of the EWS trajectory. A positive value of $2a$ indicates an upward acceleration in EWS scores, which may reflect rapid clinical deterioration.

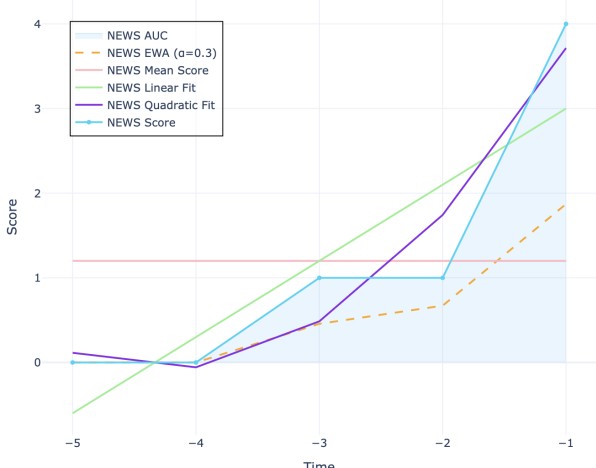

Fig. 1. Illustration of a 5-point preintervention window for a single mission using the NEWS score.

TABLE I
ONE-SAMPLE T-TEST $p$-VALUES AND COHEN'S $d$ FOR SLOPE FEATURES
TESTING FOR POSITIVE PRE-INTERVENTION TREND.
* $p < 0.05$, ** $p < 0.01$, *** $p < 0.001$.

| EWS Model | Slope $p$-value | Cohen's $d$ |
|---|---|---|
| NEWS | 9.49e-7*** | 0.0720 |
| ViEWS | 9.52e-5*** | 0.0563 |
| CEWS | 8.57e-20*** | 0.1369 |
| SEWS | 7.47e-9*** | 0.0856 |
| Worthing PSS | 0.9796 | -0.0309 |
| MEWS | 0.3485 | 0.0059 |
| GMEWS | 6.87e-7*** | 0.0729 |
| RAPS | 5.28e-13*** | 0.1078 |

$|r| \approx 0.05$, $0.1$, $0.2$, and $0.3$ as very small, small, medium, and large effect sizes, respectively [18]. Statistical significance was defined as $p < 0.05$ for both test types.

To assess the predictive value of the derived features, we trained separate L1-regularized logistic regression models with balanced class weights for each EWS to predict ROSC prior to hospital arrival from aggregated EWS totals and individual scored vital sign parameters. Model development was conducted using the domain-agnostic AI platform proposed by Weigell et al. [19]. Data were split into training and test sets using stratified sampling with a 70/30 ratio, and all features were standardized using z-score normalization. Hyperparameters were optimized via 5-fold cross-validation. For converting probabilistic outputs to binary classifications, we used the standard threshold of $0.5$, balancing the trade-off between sensitivity and specificity. We evaluated model performance on the held-out test set using the area under the receiver operating curve (AUROC), precision, recall, and F1-score. Using the same protocol, we trained two additional baselines: an EWS Snapshot model, incorporating only the most recent parameter scores and their aggregate, and a Raw Vital Sign model, in which EWS-derived features were substituted by features directly extracted from the raw vital signs.

## III. RESULTS

We analyzed whether temporal EWS-derived features carry discriminative and predictive value for ROSC outcomes in prehospital settings. The results are structured as follows. First, we report on preintervention EWS trends using slope analysis. Then, we evaluate feature distributions across outcome groups, and finally, we present the performance of predictive models trained on the derived features.

### A. Preintervention Trends: One-Sample t-Test on Slopes

We performed one-sided, one-sample t-tests on the preintervention slopes for each EWS model to assess whether EWS trajectories increased prior to intervention. Six out of eight models (CEWS, RAPS, NEWS, GMEWS, SEWS, and ViEWS) demonstrated statistically significant positive slopes ($p < 10^{-4}$), as shown in Table I. In contrast, MEWS and Worthing PSS showed non-significant results, suggesting no consistent rise in EWS values was observable. Effect sizes (Cohen's $d$) across all models were neglectable ($|d| < 0.14$), indicating limited strength despite statistical significance.

We calculated the area under the EWS curve (AUC) as a proxy for risk accumulation, which accommodates irregularly timed vital sign measurements and captures both the magnitude and duration of elevated scores. Based on this, we derived the time-normalized mean, calculated as the AUC divided by the duration of the window. We computed the most recent Exponentially Weighted Average (EWA), which emphasizes recent changes while reducing noise; the smoothing factor $\alpha$ was set to $0.3$. The EWS variability was quantified using two metrics. The time-normalized standard deviation (STDD), which represents the window's EWS STDD, divided by the window's duration, and the time-normalized delta, which encodes the difference between the maximum and minimum EWS values, also divided by the window's duration. Finally, we extracted the most recent EWS value prior to the intervention to enable comparison between singular point estimates and temporally derived features.

### D. Evaluation: Statistical Methods and Logistic Regression

We applied a one-sided, one-sample t-test for each EWS model to assess whether slopes are significantly greater than zero, thereby evaluating whether a worsening trend is statistically present prior to intervention. We used Cohen's $d$ [15] to quantify effect size, adopting the convention that $|d| \approx 0.2$, $0.5$ and $0.8$ mark small, medium, and large effects, respectively. A positive $d$ indicates that the measured slopes were above zero, suggesting clinical deterioration before intervention.

Furthermore, we utilized Mann–Whitney $U$ tests [16] to compare the distribution of all extracted features between outcome groups (ROSC versus no-ROSC), thereby assessing their discriminative power. We calculated the rank-biserial correlation ($r$) [17] as the appropriate effect size measure. The rank-biserial correlation ranges from $-1$ to $1$, where positive values indicate that the feature values tend to be higher in patients who achieved ROSC compared to those who did not, while negative values indicate the opposite. We considered

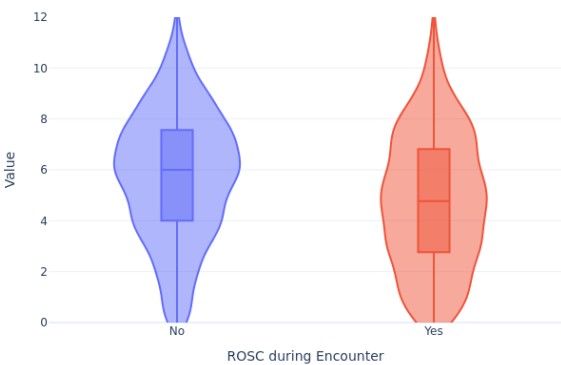

Fig. 2. Box plot comparing the preintervention mean NEWS scores between patients with and without ROSC during the encounter.

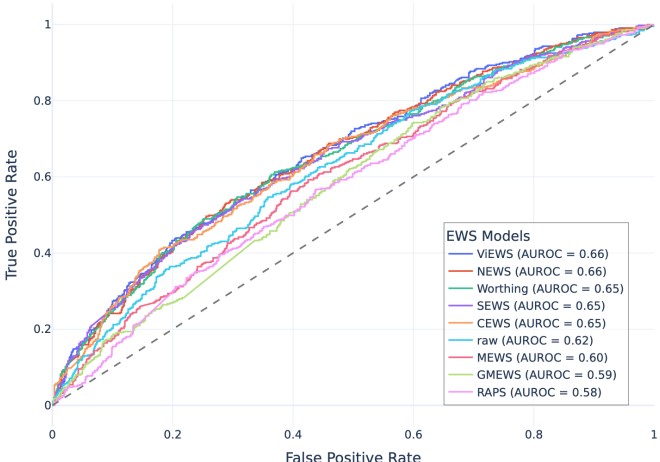

Fig. 3. ROC curves for all evaluated trend models.

## B. Feature Distributions Across Outcome Groups

We applied Mann–Whitney $U$ tests to compare feature distributions between encounters with and without ROSC prior to ED arrival, with $p$-values and rank-biserial correlation ($r$) summarized in Tables II and III. Detailed results are presented in Supplementary Material[1].

Time-normalized EWS means were significantly higher in no-ROSC cases across all models ($p < 10^{-7}$), with medium effect sizes observed for NEWS, SEWS, ViEWS, and CEWS ($|r| > 0.218$). RAPS expressed the weakest separation ($|r| < 0.1$). A representative box plot for the NEWS model is shown in Figure 2.

Temporal aggregation features, including AUC and EWA, showed strong statistical separation. Highly significant $p$-values ($p < 10^{-6}$) were obtained for all models and both features. Again, NEWS, ViEWS, CEWS, and SEWS displayed the most substantial disparity between outcome groups ($p < 10^{-25}, |r| > 0.19$).

The most recent EWS value prior to intervention differed significantly ($p < 0.04$) in all models except MEWS ($p = 0.43$). The strongest separability was found in CEWS, NEWS, CEWS, and SEWS ($p < 10^{-16}, |r| > 0.15$).

Slopes were statistically significant ($p < 0.022$) in all models besides CEWS ($p = 0.118$), effect sizes were generally very small, with $r$ ranging between 0.02 and 0.08. The most promising model was MEWS ($p = 5.04 \times 10^{-6}, r = 0.0835$). Curvature was significant ($p < 0.03$) in all models besides MEWS ($p = 0.055$) and RAPS ($p = 0.765$), however effect sizes were uniformly low ($|r| < 0.06$).

Time-normalized variability features, including STDD and delta, were the least discriminative. Among all models, only GMEWS showed statistically significant separation for both ($p < 0.002$), however with a negligible effect size ($|r| < 0.05$).

## C. Predictive Modeling via Logistic Regression

Table IV summarizes the performance metrics of the separately trained logistic regression models, which assess the predictive utility of the derived features per EWS model.

Features derived from ViEWS achieved the highest AUROC (0.665, 95% CI 0.633–0.696), followed closely by NEWS (0.659, 95% CI 0.626–0.691), Worthing PSS (0.651, 95% CIs 0.623–0.685), SEWS (0.650, 95% CIs 0.617–0.682), and CEWS (0.648, 95% CI 0.616–0.679). In contrast, features from MEWS (0.602), GMEWS (0.587), and RAPS (0.580) showed lower discriminative performance.

The EWS Snapshot models yielded lower AUROC (0.543–0.626), and the Raw Vital Sign model reached 0.625 (95% CI 0.593–0.656). More detailed performance metrics are provided in the Supplementary Material[1].

On the held-out test set, the ViEWS model achieved an overall accuracy of 63%. ROSC prediction attained a precision of 0.45, recall of 0.57, and F1-score of 0.51. No-ROSC prediction resulted in a precision and recall of 0.76 and 0.66, respectively. ROC curves of all trend models are shown in Figure 3.

## IV. DISCUSSION

Our analysis demonstrates that the temporal dynamics of EWS preceding prehospital interventions carry meaningful prognostic information related to patient outcomes.

Six of eight EWS models (NEWS, ViEWS, GMEWS, RAPS, CEWS, and SEWS) exhibited statistically significant positive preintervention slopes, confirming a consistent upward trend and deterioration prior to intervention. However, the associated effect sizes were neglectable across all EWS models, suggesting that while trends exist, their individual discriminative strength is limited.

When comparing derived features between patients with and without ROSC prior to ED arrival, the aggregated temporal features—mean, AUC, and EWA—consistently showed strong statistical separability. Overall, the largest effect sizes were detectable for NEWS, ViEWS, CEWS, and SEWS, indicating a medium effect that may be of clinically relevant use. The negative direction of $r$ across all models indicates that mean, AUC, and EWA are usually higher in patients unable to achieve ROSC, which suggests that these patients often experience sustained physiological impairment, reflected in persistently elevated EWS values. While using the most

TABLE II
MANN–WHITNEY $U$ TEST $p$-VALUES FOR PREINTERVENTION EWS FEATURES
$* p < 0.05$, $** p < 0.01$, $*** p < 0.001$.

| EWS Model | Slope | Curvature | Mean | STDD | Delta | AUC | EWA | Last Value |
|---|---|---|---|---|---|---|---|---|
| NEWS | 1.68e-4*** | 0.0078** | 7.24e-34*** | 0.5042 | 0.4681 | 2.46e-26*** | 6.99e-32*** | 2.00e-17*** |
| ViEWS | 8.29e-5*** | 0.0044** | 8.12e-34*** | 0.1831 | 0.1771 | 8.81e-26*** | 1.11e-31*** | 7.06e-17*** |
| CEWS | 0.1175 | 0.0254* | 7.42e-33*** | 0.6222 | 0.6452 | 3.21e-26*** | 1.18e-31*** | 1.42e-20*** |
| SEWS | 0.0023** | 0.0169* | 1.38e-33*** | 0.8036 | 0.8786 | 3.68e-29*** | 6.50e-32*** | 1.30e-16*** |
| Worthing | 8.31e-6*** | 0.0082** | 6.13e-28*** | 0.0014** | 0.0007*** | 7.93e-23*** | 3.23e-25*** | 6.51e-12*** |
| MEWS | 5.04e-6*** | 0.0546 | 5.47e-9*** | 0.8238 | 0.8433 | 2.94e-9*** | 5.30e-7*** | 0.4280 |
| GMEWS | 0.0051** | 0.0221* | 1.97e-9*** | 0.4976 | 0.4733 | 2.51e-12*** | 5.58e-8*** | 0.0352* |
| RAPS | 0.0217* | 0.7650 | 9.38e-8*** | 0.9085 | 0.6030 | 2.19e-9*** | 4.00e-7*** | 0.0106* |

TABLE III
RANK-BISERIAL CORRELATION ($r$) EFFECT SIZES FOR PREINTERVENTION EWS FEATURES

| EWS Model | Slope | Curvature | Mean | STDD | Delta | AUC | EWA | Last Value |
|---|---|---|---|---|---|---|---|---|
| NEWS | 0.0689 | 0.0487 | -0.2222 | 0.0122 | 0.0133 | -0.1945 | -0.2152 | -0.1545 |
| ViEWS | 0.0721 | 0.0521 | -0.2220 | 0.0244 | 0.0247 | -0.1923 | -0.2145 | -0.1519 |
| CEWS | 0.0287 | 0.0409 | -0.2187 | 0.0090 | 0.0084 | -0.1940 | -0.2144 | -0.1694 |
| SEWS | 0.0559 | 0.0437 | -0.2212 | 0.0046 | 0.0028 | -0.2053 | -0.2153 | -0.1504 |
| Worthing | 0.0812 | 0.0484 | -0.2006 | 0.0582 | 0.0616 | -0.1801 | -0.1900 | -0.1231 |
| MEWS | 0.0835 | 0.0352 | -0.1068 | -0.0041 | -0.0036 | -0.1087 | -0.0919 | -0.0143 |
| GMEWS | 0.0510 | 0.0419 | -0.1098 | -0.0124 | -0.0131 | -0.1281 | -0.0994 | -0.0377 |
| RAPS | 0.0420 | 0.0055 | -0.0978 | -0.0021 | -0.0095 | -0.1096 | -0.0928 | -0.0461 |

TABLE IV
CLASSIFICATION PERFORMANCE METRICS OF EWS AND RAW VITAL
SIGN TREND MODELS FOR ROSC PREDICTION. FOR PRECISION, RECALL,
AND F1, VALUES ARE DISPLAYED AS ROSC (+) | NO-ROSC (−).

| EWS Model | AUROC | Precision | | Recall | | $F_1$ | |
|---|---|---|---|---|---|---|---|
| | | + | − | + | − | + | − |
| NEWS | 0.6589 | **0.45** | **0.76** | 0.59 | 0.65 | 0.51 | **0.70** |
| ViEWS | **0.6645** | **0.45** | **0.76** | 0.57 | **0.65** | 0.51 | **0.70** |
| CEWS | 0.6478 | 0.43 | 0.75 | 0.59 | 0.61 | 0.50 | 0.67 |
| SEWS | 0.6499 | 0.43 | **0.76** | 0.62 | 0.60 | **0.51** | 0.67 |
| Worthing | 0.6541 | **0.45** | **0.76** | 0.58 | 0.65 | 0.51 | **0.70** |
| MEWS | 0.6015 | 0.39 | 0.74 | 0.65 | 0.49 | 0.49 | 0.59 |
| GMEWS | 0.5865 | 0.38 | 0.74 | **0.69** | 0.43 | 0.49 | 0.55 |
| RAPS | 0.5796 | 0.37 | 0.72 | 0.68 | 0.42 | 0.48 | 0.53 |
| Raw Vital Signs | 0.6249 | 0.42 | 0.74 | 0.58 | 0.60 | 0.49 | 0.67 |

recent EWS value preceding intervention as a single snapshot retains informational value, its significance and effect sizes were weaker for all models compared to aggregated features, supporting the assumption that observing EWS trajectories provides more informative context than relying on a single point in time. Derivative-based features, including slope and quadratic curvature, illustrated significant differences between ROSC and no-ROSC outcomes. Nonetheless, very small effects were recorded, and the direction of $r$ indicates that ROSC cases showed slightly steeper upward trends than no-ROSC cases. In combination with the aggregated features, this might reflect a ceiling effect in no-ROSC cases, where already high EWS levels leave less room for further increase. These results imply that derivative-based trends may carry insights but should be combined with other features as they may underperform in isolation. Variability metrics, STDD and delta, were the least informative features, with negligible effect sizes and significance. These findings highlight that short-term variability alone contributes limited insight, especially when compared to aggregated features.

When incorporated into L1-regularized logistic regression classifiers, the derived EWS features achieved moderate predictive performance (AUROC 0.580–0.665). The ViEWS-based model achieved the highest AUROC of 0.665, followed closely by NEWS and Worthing PSS. MEWS, GMEWS, and RAPS demonstrated weaker discrimination, underscoring the influence of model selection on predictive performance in the prehospital setting. In contrast, EWS Snapshot and Raw Vital Sign models yielded lower AUROC (0.543–0.626), demonstrating that temporal EWS aggregation outperforms single-point scores, and the EWS abstraction adds discriminative signal beyond raw vital trends.

All models were evaluated on their ability to detect cases in which ROSC was achieved prior to ED arrival. Performance trends revealed consistently medium to high recall but lower precision across models. Although this pattern is desirable in terms of capturing as many recoverable patients as possible, these findings suggest that models tend to favor sensitivity over specificity for ROSC, increasing the likelihood of false-positive predictions, which in a clinical decision support setting may lead to misdirected interventions. Additionally, recall and precision were consistently higher for predicting no-ROSC cases. While this bias may help reduce alarm fatigue in clinical settings, it also raises the risk of under-detecting high-risk patients. These findings emphasize the need for careful threshold optimization and balancing sensitivity and specificity in practice.

### A. Comparison to Related Work

Notably, this work advances current research by explicitly addressing short-term on-scene outcomes, which have received less attention than longer-term endpoints such as ICU admission or 3- to 30-day mortality [8]–[10]. Additionally, as

opposed to previous studies that evaluated singular, static EWS values [8]–[10] and utilized hospital-based risk thresholds [11], our study applies and assesses temporal EWS trends, thereby building upon Veldhuis et al. [12] and capturing dynamic changes in patient status leading up to intervention. Furthermore, our approach focuses solely on the four most commonly recorded vital signs (heart rate, respiratory rate, systolic blood pressure, oxygen saturation), going for a broadly applicable, simplistic approach compared to previous studies that rely on laboratory data or additional parameters [20], [21].

### B. Implications

By confirming the informational and predictive potential of preintervention vital sign trends, we lay the groundwork for future work to model prehospital patient trajectories. Analyzing the temporal behavior of vital signs, whether through aggregated representation such as EWS or directly, holds considerable potential for EMS as it could support real-time risk assessment, improve decision uncertainty, and enable the delivery of personalized, timely care.

In a real-world EMS setting, prehospital risk assessment tools could be integrated into existing patient monitoring systems. As vital signs are continuously recorded, embedded software could compute real-time EWS values and track their evolution over short time windows. Crucially, these computations are resource-efficient and could run entirely on-device. Simple visualizations, such as trend arrows, trajectory plots, or alert flags based on selected features, could assist EMS personnel in identifying early signs of deterioration in a clinically transparent, standardized, lightweight manner.

### C. Limitations

This study has several limitations. The retrospective nature of the NEMSIS dataset inherently limits control over data collection, leading to potential documentation gaps and inconsistencies, particularly in the annotation of interventions, which may lead to bias or noise. Quantitative intervention quality metrics (e.g., CPR rate) that could confound ROSC outcomes are not captured in the NEMSIS dataset. Consequently, analyses were conducted under the assumption of average intervention quality. In addition, the irregular timing of vital sign measurements and the absence of metadata regarding vital sign acquisition limit the assessment of measurement reliability. Moreover, relying on documented ROSC outcomes as ground truth may be problematic, as the labeling methodology is not explicitly reported.

Exclusion bias may arise when encounters with missing vital signs are excluded from analysis. Future efforts will address missing data more explicitly to avoid this. Furthermore, the EWS models used in this study apply fixed parameter thresholds that do not account for individual physiological baselines. This may result in false alarms or missed deterioration in patients with atypical but stable vital signs, such as athletes or those with chronic conditions.

Moreover, the utilization of linear metrics oversimplifies the complex nature of physiological deterioration. Although

quadratic fit captures some of this complexity, other modeling approaches and more flexible models that can capture non-linear dynamics will be required to fully characterize patient health states in dynamic environments. The identified limitations highlight multiple opportunities for future research, particularly through the application of sophisticated AI methods, as detailed in Section V-B.

## V. Conclusion and Future Directions

This section summarizes our key findings and highlights directions for future research.

### A. Conclusion

This study explored the informational and predictive value of temporal EWS dynamics during prehospital cardiac arrest encounters. We identified statistically significant patterns of physiological decline prior to EMS intervention, confirming that clinical deterioration signals are detectable in prehospital settings. Temporal aggregations (mean, AUC, EWA) demonstrated strong statistical separability between patient outcomes, specifically for NEWS, ViEWS, CEWS, and SEWS ($p < 10^{-26}$, $|r| > 0.19$). In contrast, models including MEWS and RAPS exhibited weaker discrimination. In predictive modeling, ViEWS-derived features achieved the highest AUROC (0.665), though model performance across all EWS models remained moderate (AUROC: 0.580–0.665), indicating room for further refinements. Overall, these findings validate that preintervention EWS trajectories encode measurable patterns of deterioration beyond static assessments and establish a transparent benchmark for on-scene outcome prediction. Furthermore, they provide a foundation for future work aimed at real-time risk assessment in EMS settings that will ultimately enhance early recognition and improve EMS outcomes.

### B. Future Directions

Building upon these findings, several promising directions for future research and methodological advancements emerge, particularly through the application of sophisticated AI methods, which are well-suited to capture intricate, non-linear dynamics in vital sign trajectories and may reveal subtle early deterioration signals, supporting time-to-event predictions and offering a more nuanced understanding of patient deterioration processes. Importantly, the temporal features explored in this study can serve as descriptive markers and inputs for future AI-driven risk assessment tools in EMS.

Moreover, AI-based approaches can learn patient-specific baselines from early-encounter data and adjust the scoring to reflect individual contexts. Personalization would likely improve sensitivity and specificity, reducing false alarms and enhancing the clinical utility of real-time risk monitoring.

While this paper focused on short-term patient trajectories, future research could explore integrated models that combine short- and long-term predictions for holistic decision support.

Furthermore, AI approaches offer a way to exploit the prognostic value of missing data rather than simply discarding

incomplete records. Future work should characterize missing-ness patterns and integrate methods that harness it, such as advanced imputation, explicit masking of missing features, or deploying models that handle partial inputs by design.

Incorporating multi-modal data streams like ECG signals could further enhance model stability and predictive accuracy. Unlike traditional EWS frameworks requiring fixed thresholds, AI approaches can learn relevant patterns and decision boundaries directly from heterogeneous inputs, accommodating new data sources without manual reconfiguration.

Despite its promise, the integration of AI into EMS workflows faces several challenges. Ensuring explainability is a crucial task, especially in high-stakes, time-sensitive environments, as it is essential that predictions can be understood and trusted by EMS personnel [22]. Additionally, the dynamic nature of EMS environments makes model drift and robustness critical concerns [23]. Over time, shifts in clinical practice, patient demographics, or data collection routines may degrade model performance [23]. Ongoing monitoring and model updates will be necessary to ensure reliability. Generally, data quality and completeness of real-world EMS data remain ongoing challenges [24]. As with any data-driven approach, the value of AI models is directly tied to the quality of the data they are trained on. Nevertheless, if these challenges are addressed, AI-driven approaches hold substantial promise for advancing prehospital care, enabling earlier interventions, improving risk stratification, and ultimately enhancing patient survival and recovery.

## ACKNOWLEDGMENT

The content reproduced from the NEMSIS Database remains the property of the National Highway Traffic Safety Administration (NHTSA). The NHTSA is not responsible for any claims arising from works based on the original Data, Text, Tables, or Figures.

The readability of this text has been improved using AI technologies.

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
