# OpenReview forum: "Early Warning Score Trend Analysis: A Data-Driven Approach for Emergency Medical Services"
_IEEE.org/EMBS/BHI/2025/Conference — BHI 2025_

### Official Review · Reviewer_8btZ · 2025-06-25

**Confidence:** 4
**Clarity Of Writing:** good
**Clinical Significance:** good
**Methodological Novelty:** good
**Overall Rating:** 5

**Experiments And Results:**

fair

**Questions For The Authors:**

1. Can you provide a sensitivity analysis to justify the selection of a 5-point pre-intervention window, as this is a critical hyperparameter?
2. Do you attribute the performance gap (AUROC 0.660 vs. >0.80 in other studies) primarily to your choice of a simple baseline model, or do you believe it reflects a fundamental limitation of pre-intervention vital signs alone?
3. Could you quantify how omitting key clinical inputs (e.g., level of consciousness) from the standard EWS models affected their validity and performance in this prehospital context?

**Strengths:**

1. This paper tackles a unique research question - short-term, pre-intervention dynamics for predicting an on-scene outcome for prehospital care research.
2. Systematic Comparison: The evaluation of eight different EWS models is methodologically sound and enhances the generalizability of the findings.
3. The study successfully establishes a quantitative baseline demonstrating that temporal vital sign information has predictive value for on-scene outcomes.

**Summary Of The Paper:**

This retrospective study analyzed 4,394 out-of-hospital cardiac arrest (OHCA) encounters from the 2021-2023 NEMSIS database. The methodology involved calculating eight distinct Early Warning Scores (EWS) using the five vital sign measurements preceding the first intervention, from which temporal features like mean, slope, and AUC were derived. A logistic regression model was used to predict on-scene Return of Spontaneous Circulation (ROSC), achieving a moderate AUROC of 0.660. The paper concludes that EWS trends hold more prognostic value than single-point measurements and provide a foundation for future AI-driven tools.

**Weaknesses:**

1. The best-reported AUROC of 0.660 is considerably lower than the 0.80-0.95 range reported by state-of-the-art machine learning models for the same task, limiting the model's immediate clinical applicability.
2. Using simple feature summarization with a linear model does not fully capture the complex, non-linear dynamics of patient deterioration.
3. The use of forward-filling and complete-case analysis for missing data is a significant limitation that can introduce bias and ignores the potential predictive signal of "informative missingness".
4. The paper frames EWS as "interpretable," overlooking modern Explainable AI (XAI) techniques like SHAP and LIME, which can pair higher-performance "black-box" models with more granular, feature-level explanations.

---

### Official Review · Reviewer_WJnq · 2025-07-07
**Early Warning Score Trend Analysis: A Data-Driven Approach for Emergency Medical Services**

**Confidence:** 4
**Clarity Of Writing:** good
**Clinical Significance:** good
**Methodological Novelty:** fair
**Overall Rating:** 4
**Final Rating:** 6

**Experiments And Results:**

fair

**Questions For The Authors:**

How did you validate that omitting the unavailable paremeters of the EWS preserves each model's predictive intent?
What is the median/IQR duration covered by the five point window used, have you tried time-normalized features?
Could you report AUROC for baselines like raw vitals and last EWS values for the prediction task?

**Strengths:**

- Pre-hospital deterioration assessment is not often studied, relative to in-hospital settings, and focusing on ROSC is a clear approach.
- The dataset used is national, increasing the chances of being diverse and containing representative samples.
- Eight different EWS models and the effects of the derived features on their predictive power are studied.

**Summary Of The Paper:**

The authors perform a retrospective analysis from the 2021-2023 National EMS Information System of 4394 cardiac arrest encounters. They calculate 8 Early Warning Score (EWS) models using four vital signs (heart rate, systolic blood pressure, oxygen saturation and respiratory rate) dropping the rest of parameters. The proposed method takes the five measurements preceding the event and creates temporal features. Then logistic regression models are trained on cohorts of patients that had return of spontaneous circulation (ROSC) and those that didn't, showing that the features differ and running a task of ROSC prediction, achieving AUROC of 0.66 with ViEWS.

**Weaknesses:**

- EWS adaptation without justification or validation (e.g. removing body temp. in MEWS or MAP in RAPS), then using the EWS thresholds for the vital signs, may invalidate the significance of each model's thresholds.
- The window for measurements does not take into account the time between measuremements, but then comparing trends among different patients. The five points could span 2 minutes or 30 minutes, likely encoding different deterioration situations and risk changes, and masking the actual deterioration "speed/slope" which is claimed to be informative.
- The proposed feature and modified models' performance are not compared to baselines such as raw vital-sign features, or vanille EWS models for the prediction task.

---

### Official Review · Reviewer_K2z7 · 2025-07-18
**Review of Submission #22**

**Confidence:** 4
**Clarity Of Writing:** excellent
**Clinical Significance:** excellent
**Methodological Novelty:** great
**Overall Rating:** 7

**Experiments And Results:**

great

**Questions For The Authors:**

1. How feasible is the deployment of this approach in real-world EMS systems? Have the authors considered latency, computational overhead, or interface design for paramedic use?

2. Would the predictive signal improve significantly if longer observation windows were used, assuming data availability? (For example, using 10 data points instead of 5 may improve the prediction?)

**Strengths:**

1. The paper addresses the real-time recognition of patient deterioration in EMS, which is a clinically significant problem, by bridging a gap in the application of EWS to prehospital care.

2. The study design is robust, with clearly defined inclusion criteria, proper handling of missing data via forward-filling, and well-structured feature engineering and evaluation using stratified sampling and regularized models.

3. The focus on temporal trends in EWS (rather than static one-time measurements) entails a novel and meaningful shift that has practical implications for how real-time monitoring can evolve in EMS systems.

4. By adopting logistic regression in their study, the authors maintain clinical interpretability while still demonstrating respectable predictive performance.

5. The discussion on limitations and future directions is particularly thoughtful and insightful. The relevant sections provide a clear roadmap for how this line of research can be extended and improved.

**Summary Of The Paper:**

This paper investigates whether short-term trends in Early Warning Scores (EWS) calculated from prehospital vital signs can be used to predict Return of Spontaneous Circulation (ROSC) during out-of-hospital cardiac arrest (OHCA) encounters. Using a retrospective cohort of 4,394 OHCA cases from the National EMS Information System (NEMSIS) between 2021 and 2023, the authors apply eight different EWS models to compute temporal features, such as slope, curvature, area under the curve (AUC), exponentially weighted average (EWA), and others, over the five most recent measurements before EMS intervention. These features are analyzed for statistical significance and used as inputs to L1-regularized logistic regression models to assess predictive performance. The findings show that temporal EWS dynamics encode clinically meaningful patterns, particularly through aggregated features like AUC and EWA, with moderate predictive performance (AUROC up to 0.660).

**Weaknesses:**

1. The paper provides a solid self-assessment in the Limitations section, and many of the key areas for improvement are already well discussed by the authors themselves.

2. While logistic regression offers interpretability, its predictive performance remains moderate (e.g., AUROC 0.58-0.66). Exploring and complementing the analysis with more flexible models, such as gradient boosting (xgboost, lgbm) or random forests, could be a relatively simple extension that better captures the non-linear dynamics of physiological deterioration.

3. Several very short paragraphs (one to two sentences) could be structurally improved by merging them with surrounding content.

4. Including a baseline model using raw vital signs or their direct trends (e.g., HR slope, BP variability) could help assess whether EWS abstraction adds or removes predictive signal. Could the authors comment on how performance might differ if raw vital signs were used instead of EWS-derived scores? Is there a risk that abstraction through EWS could mask subtle but important signals?

---

### Official Review · Reviewer_d1sn · 2025-07-21
**On-scene risk prediction**

**Confidence:** 4
**Clarity Of Writing:** good
**Clinical Significance:** great
**Methodological Novelty:** great
**Overall Rating:** 7

**Experiments And Results:**

great

**Questions For The Authors:**

Does this paper assume the intervention quality is same across all patients? We are predicting ROSC or non-ROSC but that depends not just vital trends (like you focus on in this paper) but also EMS intervention quality?

This question is important and I will appreciate the clarification. Correct if I am wrong. If my query is valid, then confounding is needed or atleast this concern must be mentioned in paper.

**Strengths:**

1. Good step in promoting risk analysis before hospital intervention - a topic less studied.
2. Thanks for mentioning NEMSIS data. A new version also just came out for 2024 as well.

**Summary Of The Paper:**

Using NEMSIS data and various EWS scores derived from vital signs, this paper does trend analysis and correlate them with ROSC and non-ROSC cases. The paper is a good step towards risk assessment during EMS intervention stage - so this is long before hospital intervention. Authors find decent prediction and separability performance using some EWS scores.

**Weaknesses:**

1. NEMSIS is not explained to satisfactory levels. Whats the frequency of vital extraction? Can you provide some code for NEMSIS data extraction? It need not be structured or even complete code.
2. Does this paper assume the intervention quality is same across all patients? We are predicting ROSC or non-ROSC but that depends not just vital trends (like you focus on in this paper) but also EMS intervention quality?
3. All EWS scores are not explained. It is kind of black box. When we observe different trends among them, you should offer some intuition.
4. Please be careful in using AI in writing. Section V.B. should be rewritten in more compact and coherent manner